

# Simulating historical flood events at the continental-scale: observational validation of a large-scale hydrodynamic model

Oliver E. J. Wing[1,2], Andrew M. Smith[1], Michael L. Marston[3], Jeremy R. Porter[3], Mike F. Amodeo[3], Christopher C. Sampson[1], Paul D. Bates[1,2]

[1]Fathom, Bristol, United Kingdom
[2]School of Geographical Sciences, University of Bristol, Bristol, United Kingdom
[3]First Street Foundation, Brooklyn, New York, United States of America

*Correspondence to*: Oliver E. J. Wing (oliver.wing@bristol.ac.uk)

**Abstract.** Continental–global scale flood hazard models simulate design floods: theoretical flood events of a given probability.
Since they output phenomena unobservable in reality, large-scale models are typically compared to more localised engineering models to evidence their accuracy. However, both types of model may share the same biases and so not validly illustrate predictive skill. Here, we adapt an existing continental-scale design flood framework of the contiguous US to simulate historical flood events. 35 discrete events are modelled and compared to observations of flood extent, water level, and inundated buildings. Model performance was highly variable depending on the flood event chosen and validation data used.
While all events were accurately replicated in terms of flood extent, some modelled water levels deviated substantially from those measured in the field. In spite of this, the model generally replicated the observed flood events in the context of terrain data vertical accuracy, extreme discharge measurement uncertainties, and observational field data errors. This analysis highlights the continually improving fidelity of large-scale flood hazard models, yet also evidences the need for considerable advances in the accuracy of routinely collected field and high river flow data in order to interrogate flood inundation models
more comprehensively.

## 1 Introduction

The severity of riverine flood hazards is principally understood through inundation modelling. Few stretches of river contain enough observations of their flood behaviour to adequately characterise the hazard they pose alone. Instead, these limited observations are used to drive physical models to produce synthetic realisations of flooding. The output of these models is
typically a flood map with a defined probability of occurrence which, when intersected with socio-economic data, can be used to estimate the frequency at which people and property may be exposed to flood hazards. Such models form the cornerstone of national flood risk management frameworks, which guide planning decisions and investment in mitigatory actions.

The gold-standard approach to building accurate flood models locally is from the ground-up by hydraulic engineers, principally
using *in situ* river flow measurements, surveyed channel bathymetry, high-resolution terrain data, and the incorporation of





local drainage and protection features. Scaling this local modelling approach up to obtain nationwide views of flood hazard therefore requires the building of many thousands of hydraulic models covering every river basin in the country. Even for the world's wealthiest countries, this poses a formidable modelling challenge. The flood mapping programme of the US Federal Emergency Management Agency (FEMA), for instance, has required over $10bn of public funding over ~50 years, yet has

only modelled one-third of US river reaches to date (Association of State Floodplain Managers, 2020).

In response to this dearth of flood hazard information at large spatial scales, geographers have built hydraulic models with domains covering vast regions or even the globe (Alfieri et al., 2014; Dottori et al., 2016; Hattermann et al., 2018; Sampson et al., 2015; Wing et al., 2017; Winsemius et al., 2013; Yamazaki et al., 2011). These models sacrifice some local accuracy

compared to the traditional engineering approach but benefit from complete spatial coverage and the ability to be re-run as climatic and landscape conditions change, all within reasonable timescale and resource limits. One of the remaining questions in this field of enquiry is: how much local accuracy is lost?

To answer it, these large-scale inundation models must be validated, but two critical barriers prevent this taking place routinely

and rigorously. Firstly, design flood maps of this nature do not represent something observable in reality. The 100-year flood, for instance, is not a tangible phenomenon since real flood events do not have spatially static return periods. In producing something theoretical, it is impossible to validate it against something real. Model-to-model comparisons – where one model is deemed to be suitably accurate so as to be the benchmark, while the other is the one to be tested – are therefore necessitated. The second barrier, then, is the low availability of suitable model benchmarks. Global flood models have been compared to

local engineering flood maps in Europe and the US, but only for a small handful of river basins, inhibiting wide-area testing (Dottori et al., 2016; Sampson et al., 2015; Ward et al., 2017; Winsemius et al., 2016). Wing et al. (2017) presented a model of the contiguous US, adopting the higher quality hydrographic, hydrometric, terrain, and protection data available in the US compared to available data globally. They compared their model to FEMA's large, yet incomplete, database of 100-year flood maps, charting a convergence of skill between the large-scale model and the engineering approach espoused by FEMA. Wing

et al. (2019) furthered this examination with statewide engineering models from the Iowa Flood Center, coming to similar conclusions. While these studies provide useful indications of large-scale model accuracy, they are fundamentally limited in their characterisation of skill through model intercomparisons. The benchmark data in these analyses may share many of the same biases (e.g., friction parameterisation, channel schematisation, structural error, terrain data precision, boundary condition characterisation) as the model being tested, and so not usefully describe the extent to which it is behavioural.


Model validation, rather than model intercomparison, can only be executed through benchmarking against observations. To do so, the hydraulic models must replicate real-world events rather than frequency-based flood maps. This would, by proxy, enable typical applications of design flood maps (such as planning or regulatory decisions, insurance pricing, or emergency response) generated by such a model to be carried out with a richer understanding of its biases. This is common practice in





event-replicating, local-scale, engineering-grade inundation modelling studies. However, their limited spatial scale, laborious
      manual set-up, and the scarce availability of validation data results in observational benchmarking against only a handful of
      real-world flood events at most – indeed, a single test case is typical (e.g. Hall et al., 2005; Hunter et al., 2008; Mason et al.,
      2003; Matgen et al., 2007; Neal et al., 2009; Pappenberger et al., 2006; Schumann et al., 2011; Stephens et al., 2012; Wood et
      al., 2016). An analysis of simulation performance across a wider variety of temporal and spatial settings would provide a more

reliable evidence case of model validity. To practically achieve this, it is necessary to replace the onerous manual construction
      and parameterisation of local-scale models with a consistent regional- to global-scale model-building framework capable of
      deployment for any model domain within its realm.

      Furthermore, the replication of historical flood events has value beyond scientific validation. While design flood maps are

useful for skilled practitioners who (mostly) understand what the models purport to represent, the maps can seem abstract and
      unconvincing to members of the public since they simulate something intangible and theoretical – requiring knowledge of the
      statistical meaning and uncertain derivation of a design flood to correctly comprehend (Bell and Tobin, 2007; Luke et al.,
      2018; Sanders et al., 2020). Producing flood maps of historical events – providing an explicit understanding of where has
      flooded in the past – can aid in motivating private mitigation efforts where the risk perception formed via a design flood map

often fails to do so (Bubeck et al., 2012; Kousky, 2017; Luke et al., 2018; Poussin et al., 2014). As such, *en masse* replication
      of historical flood events at high-resolution may have value in enhancing risk awareness and the willingness of individuals to
      take mitigatory action.

      In this paper, we adapt the existing continental-scale design flood model framework of Bates et al. (2020) to replicate historical

flood events across the contiguous US. Flood events are isolated from US Geological Survey (USGS) river gauge data, which
      form inflow boundary conditions to a ~30 m resolution 2D hydrodynamic model. High water mark surveys from the USGS
      were sourced for nine of the simulated events which, alongside derived flood extents, were used to validate the model.
      Insurance and assistance claims were obtained for 35 flood events to further analyse model skill in the context of exposure.
      By validating against an order of magnitude greater number of historical flood event observations than exist in academic

literature to date, the analysis provides robust evidence of large-scale model skill for the first time. To aid in enhancing risk
      awareness amongst the US public, these flood event footprints have been released on https://floodfactor.com/, a free and
      accessible tool for Americans to understand their flood risk. Section 2 describes the methods behind the event replication
      model and the validation procedures undertaken. In Sect. 3, the results of the model validation are presented and discussed.
      Conclusions are drawn in Sect. 4.



## 2 Methods

### 2.1 USGS Gauge Input

USGS river gauge data were initially filtered into those representing catchments which have an upstream area of $>10{,}000\ km^2$ and a record length of $>50$ years. Of these, one gauge was selected for each level 8 USGS Hydrologic Unit and the largest event in the record was extracted. This filtering ensured large loss-driving events were captured, for which validation data were more likely to be available and whose return periods could be more robustly estimated. This yielded roughly 200 river gauge events, 50 of which had suitable validation data (see Sect. 2.3). Model domains of a 50 x 50 km area were constructed around these 50 'seed' gauges and, within each domain, all USGS river gauges within it were selected (regardless of drainage area or record length) to ensure all gauged event flows were captured. 7-day hydrographs were then extracted from each gauge for each event, with the seed gauge peak at its temporal centre. To account for uncertainty due to stage measurement error and rating curve configuration in the generation of the gauged discharges (Coxon et al., 2015; Di Baldassarre and Montanari, 2009; McMillan et al., 2012), we simulate each event three times: the reported discharges as well as ±20%, producing a 0.8*Q, 1.0*Q, and 1.2*Q model for each event. Some of these 50 gauge events were of the same (particularly widespread) flood event. Once simultaneous events were merged post-simulation, 35 discrete flood events remained. Figure 1 illustrates the location of the events, with additional information provided in Table A1.

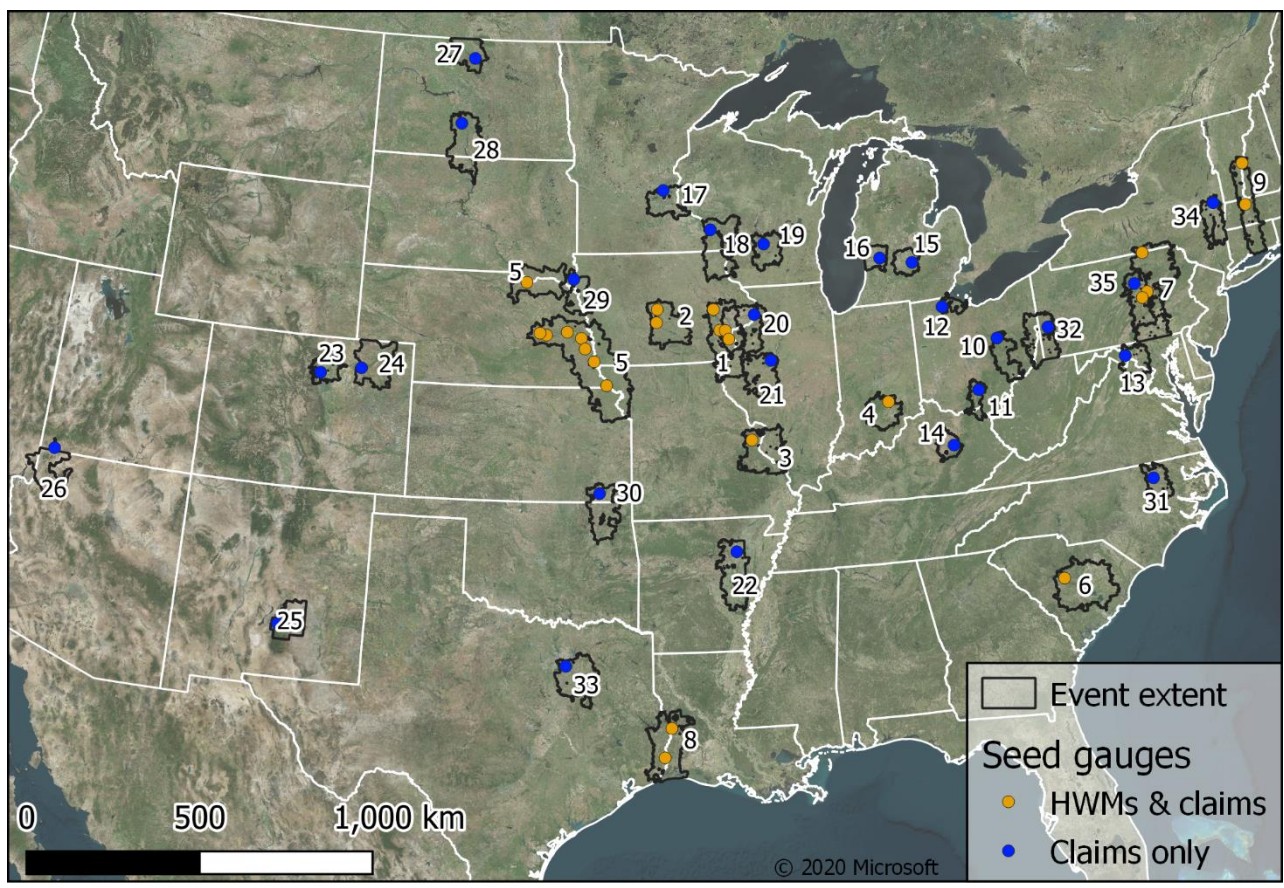

**Figure 1. Geographic distribution of 35 simulated events across the contiguous US. Coloured points represent the seed gauges and the available validation data; black outlines show the discrete event boundaries and associated IDs.**

## 2.2 Hydraulic Model

The USGS river flows form the input to the First Street Foundation National Flood Model (FSF-NFM) built in collaboration with Fathom. The model was first presented by Wing et al. (2017) with updates specified in Bates et al. (2020), based on the original global modelling framework of Sampson et al. (2015). Terrain data are based on the USGS National Elevation Dataset (NED), with hydraulic simulations run at the native resolution of 1 arc second (~30 m). Other local sources of more accurate terrain data were also compiled into the dataset where available, with ~30 m water surfaces downscaled to ~3 m in locations

where such fine-resolution data are present. Gesch et al. (2014) report relative vertical errors (point-to-point accuracy; measuring random errors exclusive of systematic errors), a more relevant control on inland flood inundation modelling accuracy than absolute vertical errors, in the NED of 1.19 m. Most, but not all, events were located in areas where the NED consists of lidar, with an associated point-to-point accuracy of ~0.66 m (Gesch et al., 2014). The computational hydraulic engine is based on LISFLOOD-FP, which solves a local inertial formulation of the shallow water equations in two dimensions

(Bates et al., 2010). River flows are routed through channels defined by the USGS National Hydrography Dataset. Channels





are retained as one-dimensional sub-grid features, permitting river widths narrower than the grid resolution (Neal et al., 2012). Channel bathymetry is estimated based on the assumption that they can convey the 2-year discharge, as estimated by a regionalised flood frequency analysis (RFFA) based on Smith et al. (2015) but using USGS river gauges (Bates et al., 2020). Ungauged river channels within each model domain propagated the RFFA-derived mean annual flow instead. Flood protection

measures are implemented directly into the model using a database of adaptations compiled from the US Army Corps of Engineers National Levee Database and hundreds of other sub-national databases. For further information, including details on the aforementioned model-to-model validation, the reader is referred to Wing et al. (2017) and Bates et al. (2020).

### 2.3 Event Validation

High water marks (HWMs) were obtained for nine of the simulated events from the USGS Flood Event Viewer

(https://stn.wim.usgs.gov/FEV/). HWMs were filtered based on: (i) the presence of a nearby upstream river gauge, ensuring the model was only tested on floodplains it simulated; (ii) a designation of being high quality; and (iii) having a vertical datum of NAVD88, consistent with the model terrain data. The number of HWMs retained for each event is shown in Table A1. The surveyed water surface elevation (WSE) from each flood event was compared to that of the nearest inundated pixel of the modelled maximum inundation extent. Performance was summarised using the following simple equations:

$$Error = \frac{\sum_1^N |WSE_{mod} - WSE_{obs}|}{N},$$  (1)

$$Bias = \frac{\sum_1^N (WSE_{mod} - WSE_{obs})}{N},$$  (2)

where $N$ is the number of HWMs and the subscripts $mod$ and $obs$ represent modelled and observed WSEs respectively. Error indicates the absolute deviation of the modelled WSE from the observed WSE: "on average, what is the magnitude of model prediction error?" Bias illustrates whether the modelled WSEs are generally higher or lower than the observed WSE: "on

average, does the model over or underpredict WSEs?"

To examine model skill in the context of flood extent prediction, HWMs are converted to maps of flood inundation in line with the methods of Watson et al. (2018). Firstly, the HWMs are interpolated to produce a 2D surface of WSEs across the model domain. These are then subtracted from terrain data, resulting in a grid of water depths. Values greater than 0 m therefore

fall within the flood extent. Areas of flooding disconnected from the river channel are removed. To prevent the generation of flood extents in areas for which there are no relevant HWMs, inundation maps are only produced in river basins (based on level 12 USGS Hydrologic Units) which contain at least one HWM. These observation-based flood extents are then compared with the extents simulated by the model using the Critical Success Index (CSI):

$$CSI = \frac{M_1 O_1}{M_1 O_1 + M_1 O_0 + M_0 O_1},$$  (3)




where *M* and *O* refer to modelled and observed pixels respectively and the subscripts 1 and 0 indicate whether these pixels are
wet or dry respectively. This metric divides the number of correctly wet pixels by the number of pixels which are wet in either
modelled or observed data. This general fit score, falling between 0 and 1, accounts for both over- and under-prediction errors.

Beyond purely flood hazard based validation, we sourced counts of buildings which were inundated during the simulated
events. Individual Assistance (IA) and National Flood Insurance Program (NFIP) claims data were gathered from the
OpenFEMA database (https://www.fema.gov/openfema). For each event, the zipcodes that intersected the event inundation
layer were selected, and the IA and NFIP claims data for those zipcodes were extracted from the claims datasets for the year
of the simulated event. The total number of claims (IA + NFIP) were then computed for each of those zipcodes. Meanwhile,
the number of building centroids inundated was computed for each zipcode using Microsoft building footprint data. Simple
statistical summaries of the errors are reported, including the coefficient of determination ($R^2$):

$$R^2 = \frac{\sum_1^N (C_{obs} - C_{mod})^2}{\sum_1^N (C_{obs} - \overline{C_{obs}})^2}, \tag{4}$$

where *C* is the count of inundated buildings observed ($_{obs}$) in OpenFEMA data or simulated by the model ($_{mod}$) across $N = 35$
events. This metric, bounded between $-\infty$ and 1, indicates the predictive capabilities of the model through comparing the
residual variance with the data variance. A perfect model would obtain an $R^2$ of 1, while (subjectively) acceptable models
would obtain $R^2 > 0.5$.

**3 Results and Discussion**

The results of the HWM validation are shown in Table 1 and visualised in Fig. 2. Biases (Eq. (2)) consistently indicate a
tendency toward underprediction for most events, even when simulated using 120% of the gauged discharge. Taking the least
biased of each event's three simulations, the mean bias comes to -0.17 m, ranging from -2.95 m for event 3 in Missouri (2015)
to 0.89 m for event 6 in South Carolina (2015) – including a simulation of event 4 in Indiana (2008) with -0.08 m. Computing
errors in line with Eq. (1), which averages the absolute deviation from the observed water surface elevation, the most accurate
of each event's simulations ranges from 0.31 m (event 4) to 2.95 m (event 3) with a mean of 0.96 m. Most of the events obtain
errors in line with the relative vertical accuracy of the NED, which is accurate to between 0.66 and 1.19 m depending on the
terrain data source (Gesch et al., 2014).






| ID | Bias (m) | | | Error (m) | | |
|---|---|---|---|---|---|---|
| | 0.8*Q | 1.0*Q | 1.2*Q | 0.8*Q | 1.0*Q | 1.2*Q |
| 1 | -0.87 | -0.55 | -0.17 | 1.31 | 1.13 | 0.96 |
| 2 | -0.61 | -0.39 | -0.10 | 0.71 | 0.53 | 0.39 |
| 3 | -4.27 | -3.75 | -2.95 | 4.27 | 3.75 | 2.95 |
| 4 | -0.65 | -0.42 | -0.08 | 0.65 | 0.45 | 0.31 |
| 5 | -1.53 | -1.11 | -0.70 | 1.53 | 1.12 | 0.74 |
| 6 | 0.89 | 1.44 | 2.07 | 1.50 | 1.84 | 2.35 |
| 7 | -1.31 | -0.94 | -0.60 | 1.96 | 1.85 | 1.85 |
| 8 | -1.28 | -0.93 | -0.59 | 1.28 | 0.97 | 0.66 |
| 9 | 0.29 | 1.05 | 1.58 | 1.08 | 1.22 | 1.70 |
| MEAN | -0.17 | | | 0.96 | | |

**Table 1. Results of the benchmarking of nine events against surveyed high water marks.**


Figure 2. Boxplots of water surface elevation errors for each of the nine simulations. 25th and 75th percentiles bound the shaded boxes with medians within. Whiskers are set to a maximum length of 1.5x the interquartile range beyond the upper or lower





**quartiles, with outliers shown as black dots. Box shading refers to the model discharge input: 80% (violet), 100% (green), and 120%**
**(orange) of the gauged discharge.**

Surveyed water marks are an excellent tool for validating inundation models, though are not themselves error-free. Numerous past studies sought to quantify uncertainties in these observational data, finding average vertical errors in the region of 0.3–0.5 m, though in some cases can be much higher and systematically biased for particular sites (Fewtrell et al., 2011; Horritt et al., 2010; Neal et al., 2009; Schumann et al., 2007). Given these constraints, typical reach-scale hydrodynamic models of
inundation events are calibrated to <0.4 m deviation from observations of water surface elevation (Adams et al., 2018; Apel et al., 2009; Bermúdez et al., 2017; Fleischmann et al., 2019; Matgen et al., 2007; Mignot et al., 2006; Pappenberger et al., 2006; Stephens and Bates, 2015; Rudorff et al., 2014). Commonly, calibration of such models is executed via maximising some measure of fit to benchmark data by varying the friction parameters (e.g. Pappenberger et al., 2005). Equally, studies have calibrated models by varying other uncertain model features, including: channel geometry (e.g. Schumann et al., 2013), terrain
data (e.g. Hawker et al., 2018), model structure (e.g. Neal et al., 2011), or boundary conditions (e.g. Bermúdez et al., 2017). The model in this study is essentially calibrated by varying the uncertain boundary conditions – though with a sparser exploration of parameter space (only three simulations per event) than is typical – to within similar errors found in the literature for some events, though most events have significantly higher errors. The impact of discharge uncertainty is evident in the errors of each simulation per event. Assuming ±20% error in the observation of flood peak stage and its translation to discharge
(a modest assessment of their uncertainties), hydraulic model errors can increase by between 6 and 107% (median of 57%). While this illustrates considerable sensitivity, different input discharge configurations within these uncertainty bounds failed to induce inundation model replication of the HWM elevations for most events.

To further contextualise the results obtained here, we analyse the hydraulic plausibility of the surveyed HWMs along select
river reaches. Figure 3a shows the profile of water surface elevation experienced during the 2008 event on the Cedar River. Figure 3c shows the same but for the Platte River event in 2019. It is clear that the HWMs produce some local water surfaces which qualitatively appear inconsistent and implausible at the reach scale. No hydraulic model obeying mass and momentum conservation laws could feasibly reproduce such water surfaces. For these events, local USGS river gauges are obtained (Cedar River: 05453520, 05464000; Platte River: 06796000, 06801000, 06805500) and a water surface for the considered reaches is
linearly interpolated between these. While being an unreliable estimate of WSE far from gauged locations, this interpolated surface provides a useful indicator of how HWM WSEs vary across the river reaches in Fig. 3b and 3d. Altenau et al. (2017a) measured water surface elevations at ~100 m intervals along a 90 m reach of the Tanana River, AK using airborne radar. The radar data, shown to be highly accurate (±0.1 m) when compared to field measurements, illustrated a smooth and approximately linear slope, even for a complex braided river. While this campaign did not take place during a flood event, it does lead one to
question whether the slopes purported by the HWMs in Fig. 3 are physically realistic. When data points in Fig. 3 are restricted to within 1 km of gauge locations, the HWMs deviate from the interpolated surface by 0.79 m (Cedar) and 0.94 m (Platte) on





average. These observational data, then, may have higher errors than similar data reported in the wider literature, which provides useful context for the 0.96 m mean error obtained by the model here.


**Figure 3. Water surface profiles based on gauge and high water mark data for (a)(b) Cedar River between Cedar Falls and Waterloo, Iowa (event 1) and (c)(d) Platte River near Omaha, Nebraska (event 5). Profiles in (a) and (c) are referenced to mean sea level, while those in (b) and (d) adopt the interpolated gauged water surface as the vertical datum.**

When examining the differences between the simulated maximum flood extents and those produced from interpolating the HWMs over relevant river basins in the terrain data, the model obtains a CSI of 0.87 on average (see Table 2). Event 3 in
Missouri obtains the lowest maximum CSI of 0.82, while the highest of 0.94 is held by event 1 in Iowa. Optimum simulations and their comparison to the observation-based extents are shown in Fig. 4.

| ID | Critical Success Index | | |
|---|---|---|---|
| | 0.8*Q | 1.0*Q | 1.2*Q |
| 1 | 0.90 | 0.92 | 0.94 |
| 2 | 0.84 | 0.88 | 0.91 |
| 3 | 0.59 | 0.70 | 0.82 |
| 4 | 0.78 | 0.82 | 0.86 |
| 5 | 0.73 | 0.80 | 0.85 |
| 6 | 0.85 | 0.86 | 0.87 |
| 7 | 0.81 | 0.82 | 0.83 |
| 8 | 0.81 | 0.85 | 0.88 |
| 9 | 0.59 | 0.88 | 0.85 |
| MEAN | 0.87 | | |

**Table 2. Results of the benchmarking of nine events against HWM-derived flood extents.**

Legend:
- True Positive
- False Positive
- False Negative
- HUC 12

Panels: (a) ID = 1, (b) ID = 2, (c) ID = 3, (d) ID = 4, (e) ID = 5, (f) ID = 6, (g) ID = 7, (h) ID = 8, (i) ID = 9

© 2020 Microsoft



**Figure 4. Maps illustrating the similarity between modelled flood events and HWM-derived extents in relevant HUC 12 zones. Interior tick marks are spaced 0.25° (~27 km) apart. (Imagery © 2020 Microsoft)**

Typical reach-scale 2D inundation models generally obtain CSIs of 0.7–0.8 when calibrated to air- or space-borne imagery of flood extents (Aronica et al., 2002; Di Baldassarre et al., 2009; Horritt and Bates, 2002; Pappenberger et al., 2007; Stephens and Bates, 2015; Wood et al., 2016). Where observations and their classification methods are high quality, CSIs of up to 0.9

can be obtained (Altenau et al., 2017b; Bermúdez et al., 2017; Bates et al., 2006; Stephens et al., 2012). The model here obtained CSIs of 0.8–0.9, though when benchmarked against interpolated HWMs rather than actual two-dimensional observations of flood extent. While it is likely that the model has replicated the uncertain benchmark flood extents within error, setting precedent for CSIs in the wider literature can often be misinformative. In Fig. 5, the relationship between minimum WSE error and maximum CSI for each of the nine events is compared. While a generally intuitive negative relationship

between CSI and water level error is exhibited (Pearson's r = -0.6), CSIs do not drastically reduce as water level errors increase. Reasons for this seeming CSI insensitivity relate to specific characteristics of each event. Events on the Congaree and Connecticut Rivers (events 6 & 9) have fairly large water level errors in spite of excellent CSIs. This is because, in both modelled and observed floods, the floodplain was filled up, meaning extent comparisons were less sensitive to model overprediction. These events were simulated with an overprediction bias: there was too much water on the floodplain (in three

dimensions), but this made little difference to flood extent (in two dimensions). In contrast, the event on Flatrock River (event 4), which is characterised by low WSE errors, obtained a comparable CSI to events 6 & 9. While vertical errors were small, the model did not completely replicate the larger flood inundation across low-gradient terrain represented by the benchmark flood extent for event 4. Even the event on Meramec River (event 3) obtained a CSI illustrative of high performance, in spite of very high water level errors. This particular flood was large in magnitude, meaning the reward for capturing the numerous

inundated areas overshadowed the penalisation for underestimating the flood edge. In the vertical plane, however, the Meramec River event simulation is shown to significantly underpredict water surface elevation. Meanwhile, the model was rightly rewarded by both metrics for correctly simulating the Iowa 2008 floods (events 1 & 2) – large floods on large rivers – by filling up the floodplains with reasonably low water level errors. These examinations build on the evidence provided by Stephens et al. (2014), who note that similar water level errors can result in different CSIs depending on the size of the flood, the valley

gradient, and the sign of model bias. Mason et al. (2007, 2009) similarly posit that an analysis of water heights offers a way of discriminating between the equifinal model structures extent comparisons tend to result in. For this analysis of a larger sample of flood events, we reaffirm their conclusions that CSIs cannot be easily compared for events of different natures and that comparison against water levels is a more discriminatory metric.




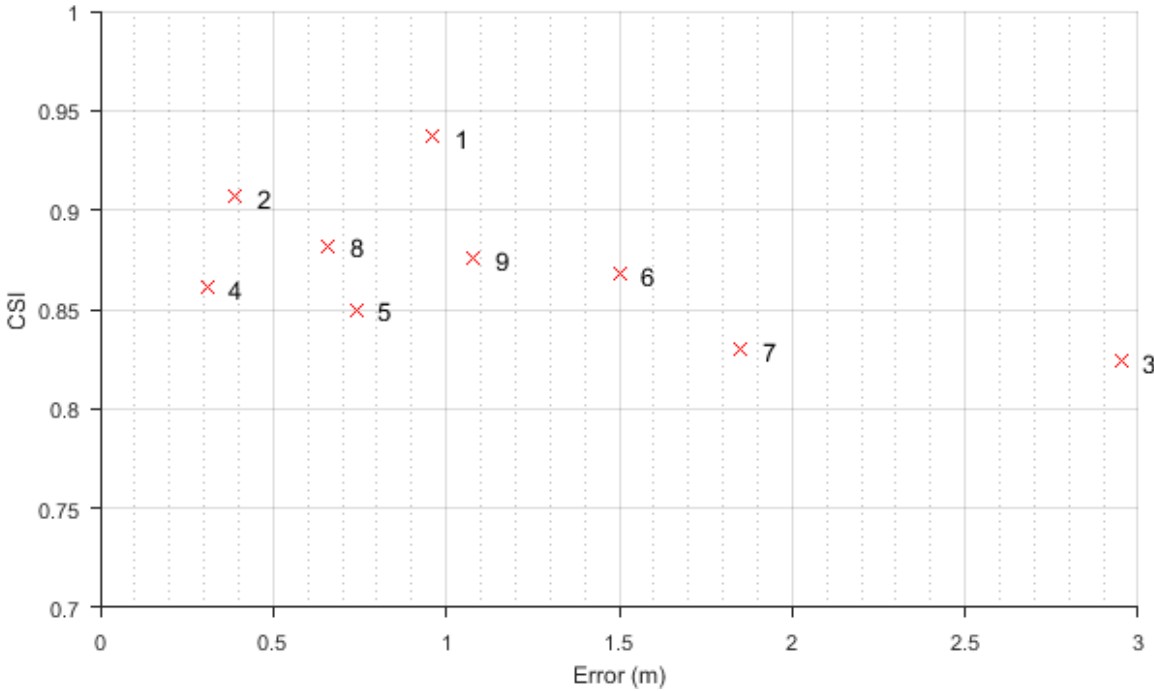

**Figure 5. The relationship between WSE error and flood extent CSI for the nine events. Numbered crosses refer to the ID of the flood event.**

The extremely large underprediction errors for the Meramec River flood event simulation (event 3) may be explained by its nature as a tributary of the Mississippi River and the arbitrary nature of the geographic domains the automated model builder produces. Even adding 20% to the reported USGS discharges resulted in an underprediction of water surface elevations by roughly 3 m. It is likely that this flood event was primarily driven by Meramec River flows backing up against the Mississippi River. With no Mississippi River gauge within the 50 km radius of the Meramec River seed gauge, the Mississippi River was not also in flood during the simulation. In the absence of a correct downstream boundary inducing a backwatering effect, the Meramec River flood freely flowed down the Mississippi River in the simulation rather than onto the Meramec River floodplain. The general underprediction bias across most flood events is likely explained by river gauge density. Boundary conditions are only available in the presence of a USGS gauging station, meaning ungauged tributaries and other lateral inflows within the model domain were not properly accounted for. Failing to simulate the aggregation of these flows, the volume of water within and exiting the model was likely much lower than reality for many of the flood events, resulting in a correspondingly underpredicting inundation model.


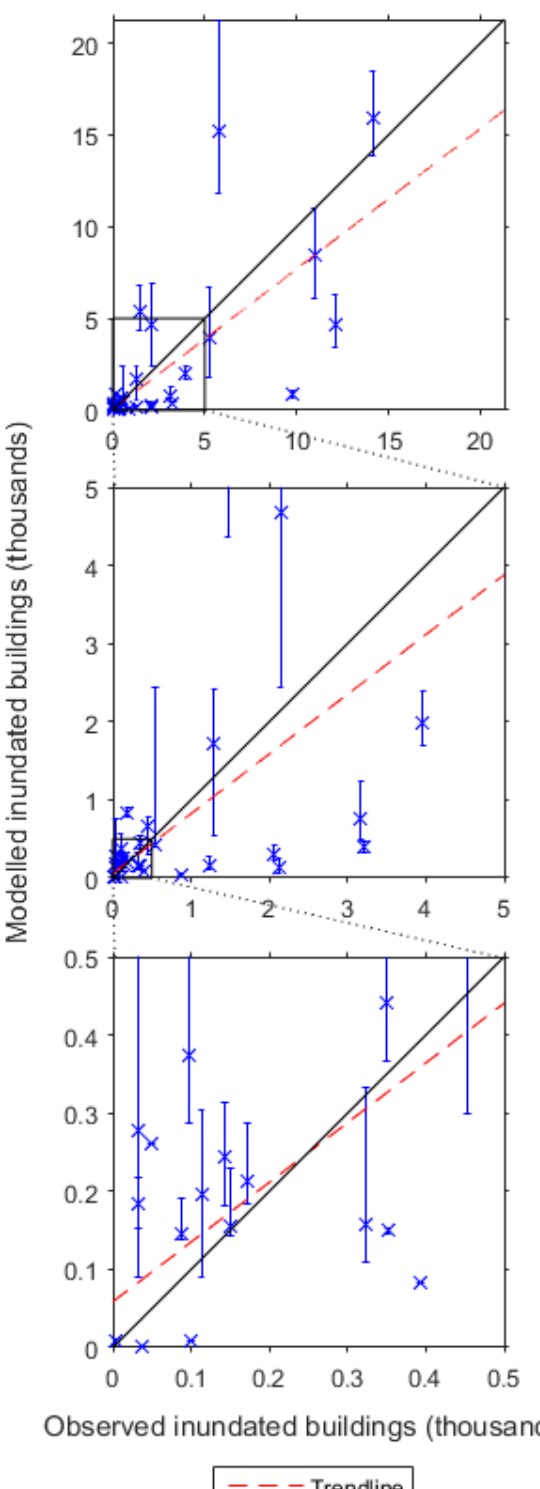





**Figure 6. Scatter plots illustrating the observed (NFIP + IA) versus the modelled count of inundated buildings for each of the 35 events. Blue crosses represent the simulated count in the 1.0*Q model, with error bars representing the range of counts between the 0.8*Q and 1.2*Q simulations. The trendline represents a linear polynomial fitted to the optimum of each event's three simulations. Descending panels are sequentially more zoomed in to the origin.**

The results using FEMA claims data to validate the full set of 35 events are shown in Fig. 6. Only 9 events simulate the correct number of claims within the discharge uncertainty bounds. When considering the closest match simulation (of the three) in terms of inundated buildings for each event, the mean error in simulated counts of building inundation is 26% of the observed count. The standard deviation of this quantity is 132%, reflecting the substantial scatter evident in Fig. 6. The modelled building inundation obtains an $R^2$ (Eq. (4)) of 0.63. In general, it appears that more catastrophic events (in terms of inundated buildings) are more frequently underpredicted by the model: 9 events underpredict building inundation compared to 3 which overpredict for events with >500 inundated buildings. Meanwhile, less catastrophic events are seemingly systematically overestimated: 10 events overpredict and 4 events underpredict for events with <500 inundated buildings. This is perhaps explained by the nature of the validation data. The sum of NFIP and IA claims may not account for all inundated buildings during an event. Impacted households may obtain assistance from local governments (with or without aid from the federal Public Assistance Program), low-interest disaster loans from the Small Business Administration, have private insurance, or simply require no external aid – none of which are captured by the sum of NFIP and IA claims. The validation data in this instance are almost certainly underestimates of the true count of affected buildings; the magnitude of this underestimation is unknown, though is likely non-negligible. 10 of the 18 events with <500 observed inundated buildings consist entirely of NFIP claims, while only 1 of the 17 events with >500 observed inundated buildings share this characteristic. This is because IA can only be claimed when the associated event is declared as a disaster by the President. Typically these are larger disasters which exceed the state or local government's capacity to respond. As such, uninsured (via the NFIP) households impacted by these smaller events (in a risk context) who are unable to claim IA will be uncounted in this analysis, as they likely received assistance from other sources. Hence, the overprediction bias for these less catastrophic events appears intuitive. Potential causes of the underprediction biases in Fig. 6 have been highlighted previously in this section: Difficulties in defining a downstream boundary may have induced a more confined flood to be modelled than reality, and low river gauge density may have resulted in some damage-causing tributary floods to remain unmodelled. As for some of the more extreme cases of underprediction, such as the flood events in South Carolina (event 6; ~10,000 observed versus ~1000 modelled inundated buildings) and Kansas (event 30; ~2000 observed versus ~400 modelled inundated buildings), these may be explained by much of the risk being pluvial driven: since the rainfall component of the Bates et al. (2020) model was not utilised here, urban, rainfall-driven, flash flooding – which contributed to many of the inundated buildings for these events – was not captured.

Furthermore, in Fig. 7 we can see that the model skill purported by the inundated building analysis bears little relation to the flood elevation and extent errors for the nine events with this information. A positive (negative) relationship would be expected in Fig. 7a (7b), yet there is no clear trend. Event 7 in Pennsylvania is one of the least skilful events in the hazard-based analysis (WSE error of 1.85 m, CSI of 0.83), yet is within discharge error of the observed inundated building count. Of course, the



Natural Hazards
and Earth System


inundated building analysis does not measure whether the correct buildings are inundated. Instead, it tests whether the same number of buildings are inundated in aggregate, which could be a fortuitous balance of type I and II errors. In spite of this, and incorporating Fig. 5 into this discussion, it is clear that the choice of model test and the spatio-temporal setting of the test holds enormous sway over how one interprets the model's efficacy. High water level errors may have little impact on a model's ability to replicate impacted buildings; equally, inundated building counts may be highly sensitive to small water level errors.

Coupled with the inconsistency of skill scores – of any metric – between flood events of different magnitudes, locations, data richness, and other characteristics, it makes obtaining an objective and generalised assessment of the US flood model employed here challenging. Layered on top of this are the likely considerable uncertainties in the data used to interrogate the model's skill: (i) errors against HWMs are often high, but from Fig. 3 we can see that these field data sometimes make little hydraulic sense – containing errors themselves perhaps approaching those obtained by the model in many instances; (ii) the resultant

extents derived from these will share their biases and, particularly in areas unconstrained by HWMs, the interpolated surface may not well represent reality; and (iii) no reliable and integrated data exist on exact counts of buildings impacted by flood events, meaning the assimilation of NFIP and IA claims used here likely underpredicts the true value of this quantity. Whether a model is deemed 'good' therefore depends on what it is simulating and for what purpose, amidst consideration of an unknown upper limit on the desired closeness of the match between the model and uncertain validation data.

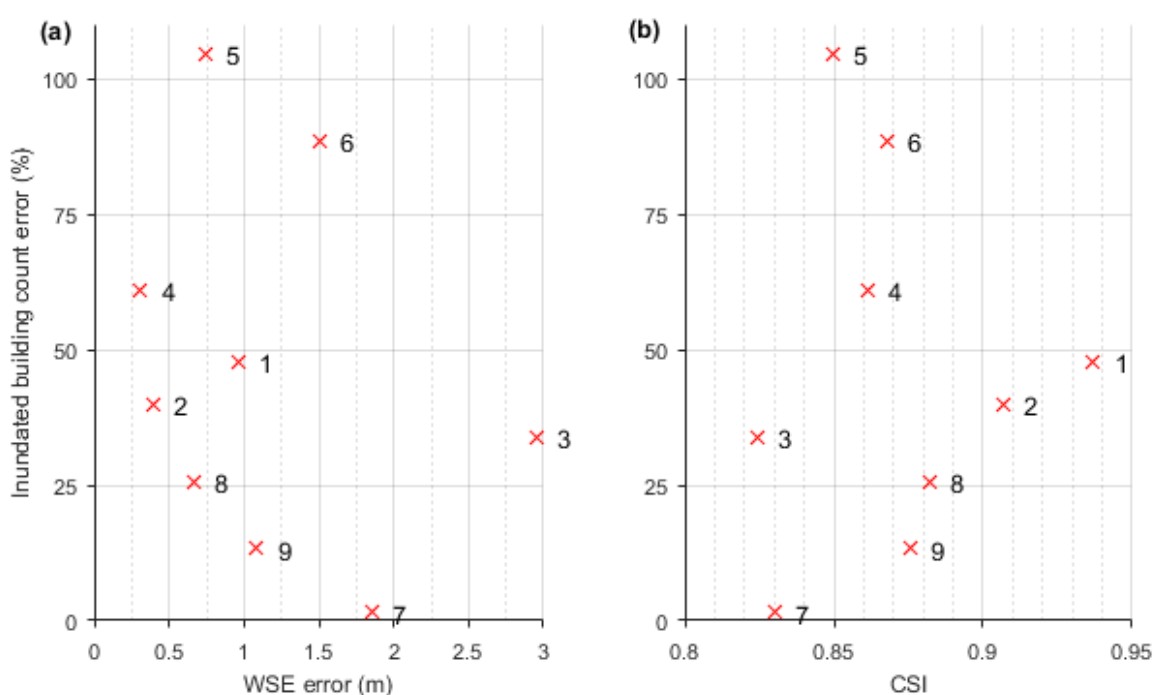


**Figure 7. The relationship between inundated building count errors and (a) WSE error and (b) flood extent CSI for the nine events. Numbered crosses refer to the ID of the flood event.**



## 4 Conclusions

In this analysis, we devised a framework to construct and deploy hydrodynamic models for any recorded historical flood event
in the US with minimal manual intervention using the continental-scale design flood model described by Wing et al. (2017)
and Bates et al. (2020). We obtained hydrologic field observations for nine events simulated by this framework and recordings
of inundated buildings for 35 such events in order to examine the skill of the model. Not unexpectedly we find model skill
varies considerably between events, suggesting that the testing of flood inundation models across a spatial scale imbalance
(i.e., benchmarking continental–global scale models against a handful of localised test cases) is prone to a misleading
evaluation of its usefulness. Previous studies suggest that the continental model employed here can replicate the extent of high-
quality, local-scale models of large flood events within error (Wing et al., 2017, 2019; Bates et al., 2020). Similarly, this
analysis illustrates the very close match between flood extents derived from field data collected during the flood events and
the maximum flood extent simulated by the continental model. However, we also highlight that tests of flood extent similarity
can mask large deviations between observed and simulated water surface elevation. While all events were well-replicated in
terms of flood extent, water surface elevation errors were roughly 1 m on average. Some events adequately replicated the
WSEs recorded in HWM data, while others were considerable underestimates. However, most event water level errors are
within the relative vertical errors of the terrain data employed. The impact of plausible (and perhaps conservative) errors in
the high flow measurements used to drive the model is shown to affect its skill substantially, yet it is also clear that other errors
remain. The insensitivity of extent comparison scores to changing water level errors suggests that CSIs are not readily
comparable for different types of flood events: a model obtaining a CSI of 0.8 for event A may be no more skilful (in terms of
water level error) than one which obtains a CSI of 0.6 for event B. We reiterate here conclusions from other bodies of work
(e.g. Mason et al., 2009; Stephens et al., 2014) which suggest that an analysis of water surface elevations provides a more
rigorous and discriminatory test of a flood inundation model. In the analysis of buildings inundated by the larger set of flood
events, some perfectly replicated the observed count of buildings while others starkly deviated from this. 26% mean error and
$R^2$ of 0.63 still indicates reasonably strong predictive skill of these quantities by the model.

Consideration of the error in the observational validation data themselves is often overlooked when interpreting the efficacy
of a flood inundation model. If the deviation between the true maximum water surface elevation achieved during a flood event
and that recorded from a high water mark is upwards of 0.5 m, obtaining model-to-observation errors of less than 0.5 m would
be the result of rewarding the replication of noise. In this analysis the magnitude of observational uncertainties is not formally
examined, yet for many of the tests the value of the validation data was close to exhaustion: that is, a given benchmark was
often replicated within its likely error. The HWMs did not always produce consistent water surfaces, interpolating between
these may produce unrealistic flood extents at some locations, and the source of inundated building data may have
undercounted the true number of impacted households.




In spite of this, useful interpretations of model performance can still be drawn from this analysis. The automated large-scale model is capable of skilfully replicating historical flood events, though in some circumstances events are poorly replicated and are generally done so with an underprediction bias. This can be addressed by further developments to the event replication framework, which include the addition of a pluvial model component and restricting event domains to those which contain

downstream (and not just upstream) river gauges to better represent backwatering. While these will be explored in future research, it is also clear from this analysis that flood inundation models can rarely be comprehensively validated using historical data. Routinely collected terrain, boundary condition, and validation data must improve drastically for the science in this field to meaningfully advance. To do this, dedicated and specialist field campaigns are required, though it should be recognised that mobilising such a resource in time to capture transient events safely during extreme floods will be challenging.

**Appendix A**

| Location | Date | ID | Seed gauges | Return Period (years) | Inundated buildings | High water marks |
|---|---|---|---|---|---|---|
| Iowa & Cedar Rivers, eastern Iowa | June, 2008 | 1 | 4 | 500 | 12,108 | 576 |
| Des Moines & Skunk Rivers, central Iowa | June/July, 2008 | 2 | 2 | 50 | 3964 | 166 |
| Meramec River, eastern Missouri | December, 2015 | 3 | 1 | 100 | 454 | 143 |
| Flatrock River, southern Indiana | June, 2008 | 4 | 1 | 75 | 3167 | 298 |
| Missouri & Platte Rivers, eastern Nebraska | March, 2019 | 5 | 8 | 100 | 5755 | 1023 |
| Congaree River, central South Carolina | October, 2015 | 6 | 1 | 20 | 9768 | 230 |
| Susquehanna River, northern Pennsylvania | September, 2011 | 7 | 3 | 150 | 14,123 | 273 |



| Sabine River, Texas/Louisiana border | March, 2016 | 8 | 2 | 1000 | 5236 | 22 |
|---|---|---|---|---|---|---|
| Connecticut River, New England | August, 2011 | 9 | 2 | 300 | 2145 | 482 |
| Killbuck Creek, eastern Ohio | January, 2005 | 10 | 1 | 25 | 393 | - |
| Scioto River, southern Ohio | January, 2005 | 11 | 1 | 5 | 351 | - |
| Maumee River, northwestern Ohio | June, 2015 | 12 | 1 | 25 | 38 | - |
| Potomac River, Maryland/West Virginia border | December, 2018 | 13 | 1 | 5 | 33 | - |
| Kentucky River, central Kentucky | May, 2004 | 14 | 1 | 25 | 172 | - |
| Grand River, central Michigan | May, 2004 | 15 | 1 | 5 | 192 | - |
| Grand River, western Michigan | April, 2013 | 16 | 1 | 15 | 97 | - |
| Mississippi River, eastern Minnesota | April, 2001 | 17 | 1 | 50 | 87 | - |
| Mississippi River, southern Minnesota | April, 2001 | 18 | 1 | 75 | 113 | - |
| Wisconsin River, central Wisconsin | June, 2008 | 19 | 1 | 5 | 871 | - |
| Mississippi River, Iowa/Illinois border | April, 2001 | 20 | 1 | 100 | 543 | - |
| Illinois River, central Illinois | April, 2013 | 21 | 1 | 75 | 150 | - |
| White River, northern Arkansas | May, 2011 | 22 | 1 | 25 | 1231 | - |





| Location | Date | | | | | |
|---|---|---|---|---|---|---|
| Boulder Creek, northern Colorado | September, 2012 | 23 | 1 | 1000 | 3212 | - |
| South Platte River, northeastern Colorado | September, 2013 | 24 | 1 | 75 | 349 | - |
| Eagle Creek, southern New Mexico | July, 2008 | 25 | 1 | 100 | 50 | - |
| Virgin River, southwestern Utah | January, 2005 | 26 | 1 | 150 | 4 | - |
| Souris River, northern North Dakota | June, 2011 | 27 | 1 | 100 | 99 | - |
| Missouri river, central North Dakota | June, 2011 | 28 | 1 | 100 | 2119 | - |
| Big Sioux River, South Dakota/Iowa border | June, 2014 | 29 | 1 | 400 | 33 | - |
| Verdigris River, southeastern Kansas | July, 2007 | 30 | 1 | 500 | 2061 | - |
| Fishing Creek, northeastern North Carolina | October, 2016 | 31 | 1 | 35 | 323 | - |
| Ohio River, western Pennsylvania | September, 2004 | 32 | 1 | 10 | 10,975 | - |
| Trinity River, northeastern Texas | May, 2015 | 33 | 1 | 50 | 143 | - |
| Hudson River, eastern New York | April, 2011 | 34 | 1 | 10 | 1290 | - |
| Susquehanna River, central Pennsylvania | September, 2004 | 35 | 1 | 25 | 1469 | - |

**Table A1. The flood events simulated in this analysis. Return periods were obtained from USGS StreamStats data.**
**Data availability**

Historical flood events were simulated on behalf of the First Street Foundation (https://firststreet.org/) and form part of their Flood Factor platform (https://floodfactor.com/). Hydrodynamic modelling output is available for non-commercial, academic research purposes only upon reasonable request from the corresponding author. USGS terrain data is available from https://ned.usgs.gov/. USGS river gauge data is available from https://waterdata.usgs.gov/nwis. USGS high water mark data are available from https://stn.wim.usgs.gov/FEV/. Insurance and assistance claims are available from https://www.fema.gov/openfema. Microsoft building footprints are available from https://github.com/microsoft/USBuildingFootprints. USGS StreamStats are available from https://streamstats.usgs.gov/. USGS HUC zones are available from https://water.usgs.gov/GIS/huc.html. The USACE National Levee Database is available from https://levees.sec.usace.army.mil/. The USGS National Hydrography Dataset is available from https://www.usgs.gov/core-science-systems/ngp/national-hydrography.

**Author contribution**

O.E.J.W performed the hydraulic analyses of the models and wrote the paper. A.M.S. and C.C.S developed and ran the models. M.L.M performed the analyses relating to the building data. J.R.P and M.F.A conceived of the project. All authors aided in the conceptualisation of the analysis and commented on initial manuscript drafts.

**Competing interests**

The authors declare that they have no conflict of interest.

**Acknowledgements**

The authors are indebted to the US Geological Survey for their continued efforts in providing easy and open access to datasets fundamental for the construction and assessment of flood inundation models in the US. Oliver Wing and Paul Bates were supported by EPSRC Grant EP/R511663/1. Paul Bates was supported by a Royal Society Wolfson Research Merit Award.

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
