# Peer review of "Simulating historical flood events at the continental-scale: observational validation of a large-scale hydrodynamic model"

_Natural Hazards and Earth System Sciences, 2020_

## Referee Comment (RC1) · Francesco Dottori (Referee) · 4 Dec 2020

This study is an valuable contribution which improves the understanding of the quality of continental-scale flood models, while highlighting the complexity of validation exercises and the existing limitations of observed data. The manuscript is well written and structured. I especially appreciated the use of a large dataset of case studies, which allows a comprehensive validation, and the detailed discussions of the different results. I think this manuscript should be accepted for publication after addressing the following issues.

- Section 3 is quite long and I would suggest splitting into subsections to improve readability

- Section 3, lines 209-24: I'm not fully convinced about the use of interpolated linear water profiles to evaluate HWMs along the two rivers. While it is true that Altenau et al. found an roughly linear water surface slope in their study, in general the water surface is influenced by the magnitude and variations of the bed slope (see Dottori et al. 2009, apologies for the self-citation). In fact, the average bed slope in the Platte River (approx 1m/km) is likely to determine a quasi-kinematic behaviour of the flood wave, with the water surface being roughly parallel to the bed profile (as indeeed suggested by the HWMs in figure 3d). Using a steady flow profile consistent with the peak flow would be more appropriate than the linear interpolation. However, I reckon that in this case simulating steady flow conditions using a 1D model would be complex in this case due to the widespread flooding, so maybe this issue could just be mentioned in the text to improve the discussion.

Minor remarks

- Line 24-25: Flood models are generally used to produce a range of flood maps, so I suggest: "The output of these models is typically one or more flood maps..."

-Line 37: "Researchers" is probably better than "geographers" here

- Section 2.1: My suggestion is to modify the title in "USGS Gauge input and event selection"

-Line 102: "Model domains of a 50x50km were constructed..." yet in Figure 1 some areas look much larger than 50x50km. Could you please explain this difference?

-Lines 172-180: I guess HWMs are referred to NED for the comparison, right?

-Lines 172-180: You could mention that the interpolated water surface map interpolated from HMWs might be affected by further errors due to the distance between HWMs

-Line 218: 90km reach

[Figure]

- Lines 370-374: Here you don't mention the possibility of combining remote sensing data (for instance flood extent from satellite imagery) with dedicated field campaigns. Given the growing availability of remote sensing data, this could be an effective strategy to improve validation datasets.

References

Dottori, F., M. L. V. Martina, and E. Todini. "A dynamic rating curve approach to indirect discharge measurement." Hydrology & Earth System Sciences 13.6 (2009).

––––––––––––––––––––

---

## Referee Comment (RC2) · Marc Bierkens (Referee) · 21 Dec 2020

Evaluation of NHESS-2020-344

This paper evaluates a continental-scale flood hazard framework by comparing the simulations with the underlying hydrodynamic model with high water marks (HWM), reconstructed flood extents and insurance claim-based estimates of affected buildings of 9 actual flooding events in the U.S. If hazard frameworks are to be trusted in reproducing correct hazard layers (inundation depth/extent for a given return period), they should be able to reproduce actual flooding events with sufficient accuracy. So, this study produces a much-needed example of performing such an evaluation. The authors are applauded for their efforts. The results are encouraging: given the errors in HWMs, the results are not that much worse in reproducing flood levels as reach-scale hydrodynamic models. Also, flood extent is reproduced with similar accuracy as reach-scale hydrodynamic models, although it was also shown that flood extent is not very discriminative between model configurations and model parameterizations.

The paper is very well written and nicely concise and should be published without much delay. I only have a small number of comments:

The most important one is the additional error that would occur if the framework would be used to estimate future inundation hazards. In that case, one cannot rely on gauged upstream water levels. Instead, input should come from hydrological models. A discussion on the errors that could be expected in that case is advised in light of the usefulness of large-scale hazard frameworks.

Line 54 "convergence of skill". Please rephrase, as it is not clear what is meant here.

Equation (1): this is commonly referred to as the "mean absolute error" and is a measure of uncertainty akin to the RMSE, but less sensitive to outliers.

Line 161: this link does not work

Equation (4): should this not be 1 – Eq. (4)?

Line 220: the realism of HWM observations is observed by comparing it with physically-realistic water levels along a reach. However, is this not a scale issue? Could local obstructions (even temporary such as debris) not have been responsible for deviating HWMs? In this case, there is not an error in the observations, but rather one has observations that has picked up local details not accounted for in the models.

Line 346: underestimates -> underestimated.

---

## Author Comment (AC1) · 4 Jan 2021

This study is an valuable contribution which improves the understanding of the quality of continental-scale flood models, while highlighting the complexity of validation exercises and the existing limitations of observed data. The manuscript is well written and structured. I especially appreciated the use of a large dataset of case studies, which allows a comprehensive validation, and the detailed discussions of the different results. I think this manuscript should be accepted for publication after addressing the following issues.

RESPONSE: We are extremely grateful to Francesco Dottori for imparting his expertise

to the benefit of this manuscript. We are delighted by his positive review; the response to his points are detailed below.

- Section 3 is quite long and I would suggest splitting into subsections to improve readability

RESPONSE: Thanks for this point. We will split this into subsections relating to the nature of the validation exercise in the revision.

- Section 3, lines 209-24: I'm not fully convinced about the use of interpolated linear water profiles to evaluate HWMs along the two rivers. While it is true that Altenau et al. found an roughly linear water surface slope in their study, in general the water surface is influenced by the magnitude and variations of the bed slope (see Dottori et al. 2009, apologies for the self-citation). In fact, the average bed slope in the Platte River (approx 1m/km) is likely to determine a quasi-kinematic behaviour of the flood wave, with the water surface being roughly parallel to the bed profile (as indeeed suggested by the HWMs in figure 3d). Using a steady flow profile consistent with the peak flow would be more appropriate than the linear interpolation. However, I reckon that in this case simulating steady flow conditions using a 1D model would be complex in this case due to the widespread flooding, so maybe this issue could just be mentioned in the text to improve the discussion.

RESPONSE: This is an interesting point and is worthy of further investigation, or further discussion at least. We will explore this in the revision.

Minor remarks - Line 24-25: Flood models are generally used to produce a range of flood maps, so I suggest: "The output of these models is typically one or more flood maps..."

RESPONSE: Thanks. Will do so.

-Line 37: "Researchers" is probably better than "geographers" here

RESPONSE: A good point: we will amend this.

- Section 2.1: My suggestion is to modify the title in "USGS Gauge input and event selection"

RESPONSE: Thank you, yes, we will modify this.

-Line 102: "Model domains of a 50x50km were constructed..." yet in Figure 1 some areas look much larger than 50x50km. Could you please explain this difference?

RESPONSE: A good point. Where events are larger than 50 x 50 km, this is where multiple 'seed gauges' are merged (as described in line 107). This means the 'final' set of events can have domains more widespread than this initial process. We will make this clearer in the revision.

-Lines 172-180: I guess HWMs are referred to NED for the comparison, right?

RESPONSE: Yes, the datums of the model and observations are consistent (NAVD88; see line 136) to facilitate a valid comparison.

-Lines 172-180: You could mention that the interpolated water surface map interpolated from HMWs might be affected by further errors due to the distance between HWMs

RESPONSE: For sure. This point only impacts the extent comparison, however: no interpolation is required for the water surface elevation comparison. We make this point in line 325 but will make it clearer here also.

-Line 218: 90km reach

RESPONSE: Thanks for spotting this.

- Lines 370-374: Here you don't mention the possibility of combining remote sensing data (for instance flood extent from satellite imagery) with dedicated field campaigns. Given the growing availability of remote sensing data, this could be an effective strategy to improve validation datasets.

RESPONSE: You're absolutely right, we'll add this to the concluding remarks.

---

## Author Comment (AC2) · 4 Jan 2021

This paper evaluates a continental-scale flood hazard framework by comparing the simulations with the underlying hydrodynamic model with high water marks (HWM), reconstructed flood extents and insurance claim-based estimates of affected buildings of 9 actual flooding events in the U.S. If hazard frameworks are to be trusted in reproducing correct hazard layers (inundation depth/extent for a given return period), they should be able to reproduce actual flooding events with sufficient accuracy. So, this study produces a much-needed example of performing such an evaluation. The authors are applauded for their efforts. The results are encouraging: given the errors in

[Figure]

HWMs, the results are not that much worse in reproducing flood levels as reach-scale hydrodynamic models. Also, flood extent is reproduced with similar accuracy as reach-scale hydrodynamic models, although it was also shown that flood extent is not very discriminative between model configurations and model parameterizations. The paper is very well written and nicely concise and should be published without much delay. I only have a small number of comments:

RESPONSE: The authors are very thankful for Marc Bierkens' review and are thrilled by his positive comments. Responses to his points are made below:

The most important one is the additional error that would occur if the framework would be used to estimate future inundation hazards. In that case, one cannot rely on gauged upstream water levels. Instead, input should come from hydrological models. A discussion on the errors that could be expected in that case is advised in light of the usefulness of large-scale hazard frameworks.

RESPONSE: This is an excellent point and we will be sure to discuss this in the revision. Not only would the use of runoff models permit exploration of future hazard, a number of the issues related to gauge density that we identify in the paper would also be ameliorated. We must also point out the loss of (present-day) accuracy induced by such an approach, however, and future research should certainly explore the impact of this.

Line 54 "convergence of skill". Please rephrase, as it is not clear what is meant here.

RESPONSE: Of course. We will make this clearer in the revision. It is meant to mean the quality of local and larger scale models is beginning to become equivalent.

Equation (1): this is commonly referred to as the "mean absolute error" and is a measure of uncertainty akin to the RMSE, but less sensitive to outliers.

RESPONSE: That's right. We'll make reference to its common reference in the revision.

Line 161: this link does not work

RESPONSE: Seems like the address has changed since preprint publication. The new one is now https://www.fema.gov/about/openfema/data-sets

Equation (4): should this not be 1 – Eq. (4)?

RESPONSE: Thanks for spotting this, you're absolutely right.

Line 220: the realism of HWM observations is observed by comparing it with physically realistic water levels along a reach. However, is this not a scale issue? Could local obstructions (even temporary such as debris) not have been responsible for deviating HWMs? In this case, there is not an error in the observations, but rather one has observations that has picked up local details not accounted for in the models.

RESPONSE: This is an excellent point. Alongside other comments, including those from Francesco, this suggestion will make for a much richer and more informative discussion.

Line 346: underestimates -> underestimated.

RESPONSE: Thanks. We will amend.

---

## Author Response (AR1)

Reviewer 1
This study is an valuable contribution which improves the understanding of the quality of continental-scale flood models, while highlighting the complexity of validation exercises and the existing limitations of observed data. The manuscript is well written and structured. I especially appreciated the use of a large dataset of case studies, which allows a comprehensive validation, and the detailed discussions of the different results. I think this manuscript should be accepted for publication after addressing the following issues.

*We are extremely grateful to Francesco Dottori for imparting his expertise to the benefit of this manuscript. We are delighted by his positive review; the response to his points are detailed below.*

- Section 3 is quite long and I would suggest splitting into subsections to improve readability

*Thanks for this point. We will split this into subsections relating to the nature of the validation exercise in the revision.*

- Section 3, lines 209-24: I'm not fully convinced about the use of interpolated linear water profiles to evaluate HWMs along the two rivers. While it is true that Altenau et al. found an roughly linear water surface slope in their study, in general the water surface is influenced by the magnitude and variations of the bed slope (see Dottori et al. 2009, apologies for the self-citation). In fact, the average bed slope in the Platte River (approx 1m/km) is likely to determine a quasi-kinematic behaviour of the flood wave, with the water surface being roughly parallel to the bed profile (as indeeed suggested by the HWMs in figure 3d). Using a steady flow profile consistent with the peak flow would be more appropriate than the linear interpolation. However, I reckon that in this case simulating steady flow conditions using a 1D model would be complex in this case due to the widespread flooding, so maybe this issue could just be mentioned in the text to improve the discussion.

*This is an interesting point and is worthy of further investigation, or further discussion at least. We will explore this in the revision. (see lines 223-231)*

Minor remarks
- Line 24-25: Flood models are generally used to produce a range of flood maps, so I suggest: "The output of these models is typically one or more flood maps..."

*Thanks. Will do so.*

-Line 37: "Researchers" is probably better than "geographers" here

*A good point: we will amend this.*

- Section 2.1: My suggestion is to modify the title in "USGS Gauge input and event selection"

*Thank you, yes, we will modify this.*

-Line 102: "Model domains of a 50x50km were constructed..." yet in Figure 1 some

areas look much larger than 50x50km. Could you please explain this difference?
*A good point. Where events are larger than 50 x 50 km, this is where multiple 'seed gauges' are merged (as described in line 107). This means the 'final' set of events can have domains more widespread than this initial process. We will make this clearer in the revision.*

-Lines 172-180: I guess HWMs are referred to NED for the comparison, right?

*Yes, the datums of the model and observations are consistent (NAVD88; see line 136) to facilitate a valid comparison.*

-Lines 172-180: You could mention that the interpolated water surface map interpolated from HMWs might be affected by further errors due to the distance between HWMs

*For sure. This point only impacts the extent comparison, however: no interpolation is required for the water surface elevation comparison. We make this point in line 325 but will make it clearer here also.*

-Line 218: 90km reach

*Thanks for spotting this.*

- Lines 370-374: Here you don't mention the possibility of combining remote sensing data (for instance flood extent from satellite imagery) with dedicated field campaigns. Given the growing availability of remote sensing data, this could be an effective strategy to improve validation datasets.

*You're absolutely right, we'll add this to the concluding remarks. (see lines 391-392)*

Reviewer 2
This paper evaluates a continental-scale flood hazard framework by comparing the simulations with the underlying hydrodynamic model with high water marks (HWM), reconstructed flood extents and insurance claim-based estimates of affected buildings of 9 actual flooding events in the U.S. If hazard frameworks are to be trusted in reproducing correct hazard layers (inundation depth/extent for a given return period), they should be able to reproduce actual flooding events with sufficient accuracy. So, this study produces a much-needed example of performing such an evaluation. The authors are applauded for their efforts. The results are encouraging: given the errors in HWMs, the results are not that much worse in reproducing flood levels as reach-scale hydrodynamic models. Also, flood extent is reproduced with similar accuracy as reachscale hydrodynamic models, although it was also shown that flood extent is not very discriminative between model configurations and model parameterizations.
The paper is very well written and nicely concise and should be published without much delay. I only have a small number of comments:

*The authors are very thankful for Marc Bierkens' review and are thrilled by his positive comments. Responses to his points are made below:*

The most important one is the additional error that would occur if the framework would be used to estimate future inundation hazards. In that case, one cannot rely on gauged

upstream water levels. Instead, input should come from hydrological models. A discussion on the errors that could be expected in that case is advised in light of the usefulness of large-scale hazard frameworks.

*This is an excellent point and we will be sure to discuss this in the revision. Not only would the use of runoff models permit exploration of future hazard, a number of the issues related to gauge density that we identify in the paper would also be ameliorated. We must also point out the loss of (present-day) accuracy induced by such an approach, however, and future research should certainly explore the impact of this. (see lines 384-387)*

Line 54 "convergence of skill". Please rephrase, as it is not clear what is meant here.

*Of course. We will make this clearer in the revision. It is meant to mean the quality of local and larger scale models is beginning to become equivalent.*

Equation (1): this is commonly referred to as the "mean absolute error" and is a measure of uncertainty akin to the RMSE, but less sensitive to outliers.

*That's right. We'll make reference to its common reference in the revision.*

Line 161: this link does not work

*Seems like the address has changed since preprint publication. The new one is now https://www.fema.gov/about/openfema/data-sets*

Equation (4): should this not be 1 – Eq. (4)?

*Thanks for spotting this, you're absolutely right.*

Line 220: the realism of HWM observations is observed by comparing it with physically realistic water levels along a reach. However, is this not a scale issue? Could local obstructions (even temporary such as debris) not have been responsible for deviating HWMs? In this case, there is not an error in the observations, but rather one has observations that has picked up local details not accounted for in the models.

*This is an excellent point. Alongside other comments, including those from Francesco, this suggestion will make for a much richer and more informative discussion. (see lines 223-231)*

Line 346: underestimates -> underestimated.

*Thanks. We will amend.*